# Detection of Venezuelan Equine Encephalitis Virus from Brain Samples of Equines with Encephalitis

**Bernal León [1,\*], Josimar Estrella-Morales [1] and Carlos Jiménez [2]**

1 Servicio Nacional de Salud Animal, Heredia 40104, Costa Rica; josimar.estrella.m@senasa.go.cr
2 Escuela de Veterinaria, Facultad Ciencias de la Salud, Universidad Nacional Campus Benjamín Núñez, Heredia 40104, Costa Rica; carlos.jimenez.sanchez@una.ac.cr
\* Correspondence: bernal.leon.r@senasa.go.cr

**Simple Summary:** In this study, we investigated the presence of Alphavirus in bovines and equines with signs of encephalitis. After comparing four Universal RT-PCR methods and selecting the one best suited for our laboratory conditions, none of the bovine brains tested positive for Alphavirus. However, out of the 30 equine samples, only four were positive using the selected Universal RT-PCR. Through sequencing, we confirmed that only two of these samples belonged to the Venezuelan equine encephalitis virus subtype IE. These samples were isolated in the northern region of Costa Rica. The presence of this virus highlights the importance of ongoing surveillance and understanding of these viruses to implement effective disease control measures under one health umbrella concept.

**Abstract:** Alphavirus species are globally distributed zoonoses primarily transmitted by arthropods. The Venezuelan equine encephalitis virus (VEEV) and Eastern equine encephalitis virus (EEEV) are endemic in Costa Rica. This study aims to detect these viruses in brain samples from equines displaying nervous signs. For this purpose, four published Universal RT-PCR methods were compared. The most sensitive and specific RT-PCR method was used to test a total of 70 brain samples, including 40 from bovines and 30 from equines, all exhibiting nervous signs. In the positive cases, eight different brain regions were extracted and tested using this RT-PCR. Positive cases were confirmed through sequencing. Torii RT-PCR demonstrated the highest sensitivity and specificity for diagnosing VEEV and EEEV/Sind among the four Universal RT-PCR assays. Not all assessed brain regions showed DNA amplification. None of the bovine brains was positive, and out of the 30 equine brain samples, only four tested positive, and sequencing confirmed two of these samples as VEEV subtype IE. Torii RT-PCR successfully detected VEEV in pools of the hippocampus, spinal cord, and basal nuclei, making these brain regions suitable for diagnosing this virus. None of the samples were positive for EEEV or WEEV.

**Keywords:** alphavirus; Costa Rica; Venezuelan equine encephalitis; RT-PCRs

## 1. Introduction

The *Alphavirus* genus comprises over 31 species, most of which are zoonotic viruses transmitted by mosquitoes [1]. Outbreaks caused by 30 of these species have had a significant impact on public health in the Americas, including diseases such as Chikungunya, Venezuelan equine encephalitis virus (VEEV), Eastern equine encephalitis virus (EEEV), and Western equine encephalitis virus (WEEV) [2]. Alphaviruses can cause a range of diseases in humans, ranging from flu-like symptoms to arthritis, rashes, and potentially fatal encephalitis [2].

Among the methods for diagnosing *Alphavirus*, polymerase chain reaction (PCR) is a reliable and sensitive technique. Various PCR variants have been modified to reduce costs while maintaining accuracy [3]. Universal PCR is one such variant that can detect virus families or genera by targeting multiple species in a single reaction [4]. Although

sequencing of the PCR product is necessary to determine the specific *Alphavirus* species or strain [5], this methodology allows for the discovery of new Alphaviruses and differentiation between various viral species affecting different hosts [6]. Several universal PCR assays have been published for diagnosing the Alphavirus genus [4,6–9]. However, given the variety of diagnostic options and potential universal primer candidates for *Alphavirus*, it is crucial to properly compare and assess the conditions and limitations of the available sets of universal primers.

In Costa Rica, VEEV and EEEV are endemic [10], while WEEV has not been detected [11]. However, VEEV is more prevalent (35.9%) (confidence interval (CI): 29.9–42.5) than EEEV (2.8%) (CI: 1.3–5.9) [10]. VEE infection in humans can manifest as fever, headache, vomiting, and diarrhea, which can sometimes be misdiagnosed as dengue fever [12]. Equines, on the other hand, may exhibit symptoms such as weight loss, ataxia, blindness, and even death [13]. This study aims to detect the presence of *Alphavirus* in brain samples collected from 2012 to 2021, including 40 cattle and 30 horses with encephalitis that tested negative for rabies. To achieve this, four Universal RT-PCR methods were compared to determine the most sensitive and specific one for detecting *Alphavirus* in these samples.

## 2. Materials and Methods

Positive Controls were provided by the Gorgas Memorial Institute of Health. The TC83 VEEV IAB strain and the EEEV/Sinv chimeric virus [14] were propagated in Vero E6 cell cultures. The monolayer of each virus was inoculated at a multiplicity of infection (MOI) of 0.1 in a 150 cm flask containing 15 mL of Dulbecco's Modified Eagle Medium (DMEM) supplemented with 2% fetal bovine serum (FBS). The flasks were incubated at 37 °C for one hour, with rocking every 15 min to ensure even distribution of the virus. After harvesting, the titers of EEE/Sinv and TC83 were determined to be $1.4 \times 10^8$ plaque-forming units per milliliter (PFU/mL) and $7 \times 10^9$ PFU/mL, respectively.

Histological analysis was performed on various brain regions that were fixed in formalin, including the spinal cord, medulla oblongata, cerebellum, frontal colliculi, thalamus, hippocampus, occipital cortex, and basal ganglia. This analysis was conducted on a brain sample obtained from an equine with encephalitis, which was collected on 22 October 2015, and had tested positive for VEEV by IgM ELISA. No fixed tissue from these locations was preserved at −80 °C for molecular analysis.

The control samples were stored at temperatures between −50 °C and −80 °C and were extracted using either the magnetic beads method (Applied Biosystems™ (Waltham, MA, USA), MagMAX™ Viral/Pathogen Nucleic Acid Isolation Kit) or the column method DNeasy Blood and Tissue Kit (Qiagen, Germantown, MD USA).

One RT-PCR and three nested and semi-nested RT-PCRs were assessed to compare the sensitivity of different Universal RT-PCR methods [6–9]. These PCR methods were evaluated for their ability to detect varying dilutions of Alphaviruses, ranging from $10^7$ PFU/mL to $10^3$ PFU/mL for the VEEV TC83 strain and from $10^6$ PFU/mL to $10^2$ PFU/mL for the EEEV/Sinv chimeric virus. The workflow for the comparison of the Universal RT-PCRs is depicted in Figure 1.

In Phase I, two-step nested RT-PCR was tested with a total of 21 dilutions of the control samples described previously. The reverse transcription step was the same for the Grywna, Pfeffer, and Sanchez protocols. The specific details of the reverse transcription master mix can be found in Table S1 of Supplementary Materials. The PCR and nested PCR conditions, including the master mix reactions and thermocycling conditions, for the Grywna, Pfeffer, and Sanchez protocols are provided in Table S2 of Supplementary Materials.

The Grywna protocols were performed using an Applied Biosystems Veriti Thermal Cycler, while other reactions without touchdown protocols were carried out in an Applied Biosystems 2720 Thermal Cycler (Waltham, MA, USA). In the Grywna protocol, a 210 bp amplicon was generated from the region of the NSP4 gene during the second round of the nested PCR. For the Pfeffer protocol, the amplicons were 434 bp and 310 bp for the first RT-PCR and the nested PCR, respectively, targeting the NSP1 gene. The Sanchez

protocol produced amplicons of 481 bp and 195 bp for each round, respectively, targeting the NSP4 gene. In all RT-PCRs, water was added as necessary to complete the master mix volume reaction. The amplicons were visualized using a 2% agarose gel stained with Gel-Red. A molecular weight ladder, specifically the 50 bp GeneRuler™ ladder, was used for size determination. The ladder contained bands ranging from 50 bp to 1000 bp, with the strongest bands corresponding to 250 bp and 500 bp.

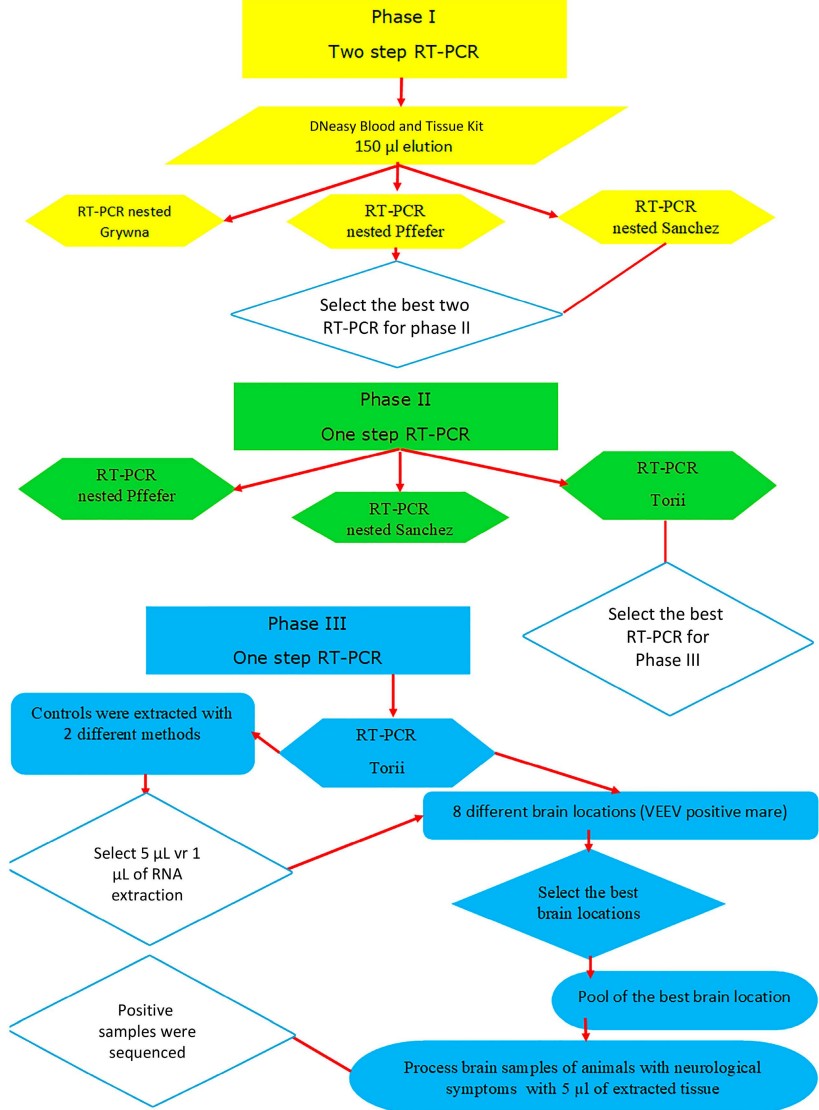

**Figure 1.** Workflow of the brain samples process.

In Phase II, the Pfeffer, Sanchez, and Torii protocols were compared in the one-step RT-PCR format. The details of these protocols can be found in Table S3 RT-PCR step of the Pfeffer, Sanchez, and Torii protocols in Supplementary Materials. Specifically, the Torii RT-PCR protocol amplified a fragment of approximately 460 bp in the NS4 gene [6]. The McNemar and Kappa tests were used to establish significant differences between the four methods evaluated during the first and second phases.

In Phase III, the comparison focused on the volume of extraction, specifically using 1 μL and 5 μL volumes. The most sensitive RT-PCR protocol from Phase II was used for this comparison. The samples used in these assays included dilutions of VEEV TC83 ($7 \times 10^5$ PFU/mL) and EEEV ($1.2 \times 10^5$ PFU/mL). Additionally, cell culture supernatants of VEEV ($5 \times 10^8$ PFU/mL) and EEEV/Sinv ($4.5 \times 10^6$ PFU/mL) were titrated on 20 August

2018 (40 months prior) and were used. These control samples were stored at −80 °C and were thawed for testing. Tenfold dilutions were prepared in DNAase and RNAse-free water, and 200 µL of each dilution was extracted using two commercial methods, columns and magnetic beads (Dneasy Blood and Tissue Kit (Qiagen, Germantown, MD, USA), magnetic beads method with Applied Biosystems™ MagMAX™ Viral/Pathogen Nucleic Acid Isolation Kit (Thermo Fisher Scientific, Waltham, MA USA), and Mag-MAX™-96 Total).

In 2015, an equine brain that tested positive for VEEV through IgM ELISA was used for further analysis. From this brain, tissue samples ranging from 53 mg to 99 mg were collected from 8 identified locations. For each sample, tissue pieces weighing between 20 mg to 25 mg were extracted using the Dneasy Blood and Tissue Kit (Qiagen, Germantown, MD, USA) and subsequently eluted in 200 µL of AE buffer. These extracted samples were then subjected to testing using the most suitable RT-PCR method determined in the study.

The brain tissue samples analyzed in the study were collected from a total of 70 animals with signs of encephalitis, including 30 equines and the remaining samples from bovines. The collection period spanned from 2012 to 2021, and all samples were stored at temperatures ranging between −20 °C and −80 °C for preservation and subsequent analysis. The extraction of nucleic acids from the brain tissue samples was performed using either the magnetic beads method with the Applied Biosystems™ MagMAX™ Viral/Pathogen Nucleic Acid Isolation Kit (Waltham, MA, USA) or the column method with the Dneasy Blood and Tissue Kit (Qiagen, Germantown, MD USA).

In the case of amplified positive samples, they were further confirmed through sequencing using the Sanger protocol. Forward and reverse primer reactions were prepared separately, with each reaction containing a 20 µL mixture comprising $0.5\times$ ready reaction premix, $0.5\times$ sequencing buffer, 8 µM forward or reverse primers, and 10 µL of DNA (10–40 ng/µL). The PCR reactions underwent thermal cycling conditions, followed by resin and SAM™ Solution purification. Depending on the band intensity observed, 10 to 20 µL of the supernatant was transferred to a 96-well plate. The sequencing step was performed using a SeqStudio genetic analyzer from Applied Biosystems™ (Waltham, MA, USA).

*Ethics*

All samples used in this study were taken according to law 8495, "General Law of the National Animal Health Service".

## 3. Results

Our primary aim was to enhance the diagnostic surveillance of animals exhibiting encephalitis signs. Throughout this investigation, we evaluated different protocol formats to determine the most effective method for our study.

### 3.1. Phase I Universal Two-Step RT-PCR Format

Figure 2 presents the results of the three nested Universal RT-PCR assays, which utilized the same control dilutions of Alphavirus. The molecular weight (MW) is a 50 bp ladder, the first five bands ranging from 50 to 250 bp, followed by bands ranging from 300 bp to 1000 bp. The strongest bands in the ladder correspond to 250 bp and 500 bp. The arrows show the amplified products in each PCR according to the dilution used. Lanes for each method correspond to the same control samples.

Figure 2A shows the Grywna nested RT-PCR results. An amplicon size of 210 bp is observed only in the dilution VEEV $10^7$ PFU/mL, indicated by an arrow; lower concentrations of VEEV or any EEEV/Sinv dilutions were not detected. In Figure 2B, the Pfeiffer nested RT-PCR exhibits positive amplicon bands at 434 bp (first RT-PCR) and 310 bp (second PCR). This method detected four more dilutions, resulting in a total of five more samples being identified compared to the Grywna test. VEEV at concentrations of $10^7$ PFU/mL and $10^5$ PFU/mL and the EEEV/Sinv controls from $10^6$ PFU/mL to two replicates of $10^4$ PFU, as was indicated by an arrow at the 434 bp band in the initial RT-PCR. However, no clear bands were observed at the 310 bp band in the nested PCR (Figure 2B). The Kappa agreement

test returned a fair agreement level of 0.22 between the two methods. Furthermore, the differences observed between these two methods were found to be statistically significant according to the McNemar test ($p = 0.0004$).

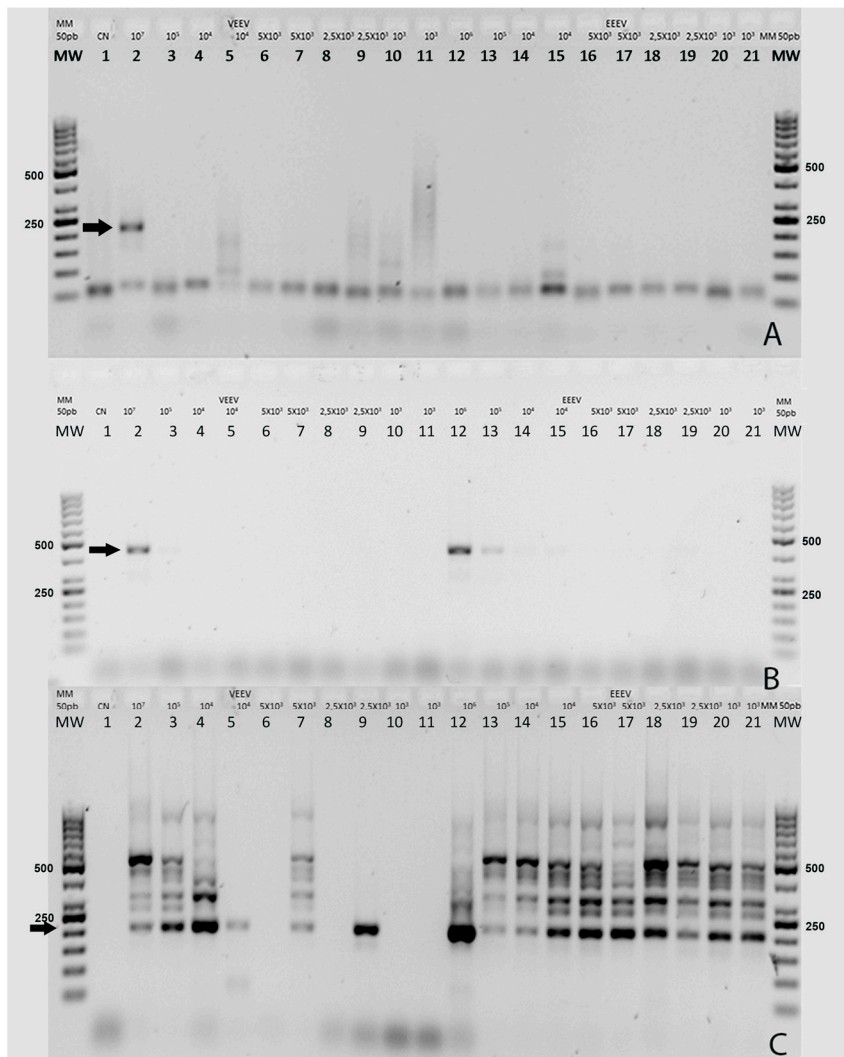

**Figure 2.** Comparison of the three universal nested RT-PCR assays using the two-step format. (**A**) depicts Grywna nested RT-PCR results, (**B**), the Pfeiffer nested RT-PCR and (**C**) shows the results of the Sanchez-nested RT-PCR. Lane 1 negative control (no virus). Lanes 2 to 11 show VEEV positive controls, with varying concentrations ($10^7$, $10^5$, $10^4$, $10^4$, $5 \times 10^3$, $5 \times 10^3$, $2.5 \times 10^3$, $2.5 \times 10^3$, $10^3$, $10^3$) expressed in PFU/mL. Lanes 12 to 21 depict the EEEV/Sinv Positive control, with the following concentrations ($10^6$, $10^5$, $10^4$, $10^4$, $5 \times 10^3$, $5 \times 10^3$, $2.5 \times 10^3$, $5 \times 10^3$, $10^3$, $10^3$) expressed in PFU/mL.

Figure 2C displays the results of the Sanchez-nested RT-PCR, showing amplicons at 481 bp (first PCR) and 195 bp (second PCR indicated by an arrow). These amplicons were consistently observed in all concentrations of EEEV/Sinv. However, it is worth noting that this band was not consistently present in all VEEV dilutions.

In comparison to the Pfeffer test, the Sanchez-nested RT-PCR detected ten more positive controls. The Kappa agreement test also showed a fair agreement level of 0.22 between the two methods. Furthermore, the differences observed were found to be statistically significant according to the McNemar test ($p = 0.0015$).

It is important to highlight that, in this assay (Figure 2C), nonspecific bands were also amplified. Specifically, the band observed in lane 13 corresponds to the EEEV/Sinv $10^3$ PFU/mL chimeric virus, which was later confirmed as Sinv through sequencing.

### 3.2. Phase II Universal One-Step RT-PCR Format

Figure 3 depicts the comparison of three one-step RT-PCR formats (Pfeffer, Sanchez, and Torii) for detecting the dilution of two viruses, VEEV and EEEV/Sinv, at various concentrations expressed in PFU/mL. The Lanes for each method correspond to the same samples.

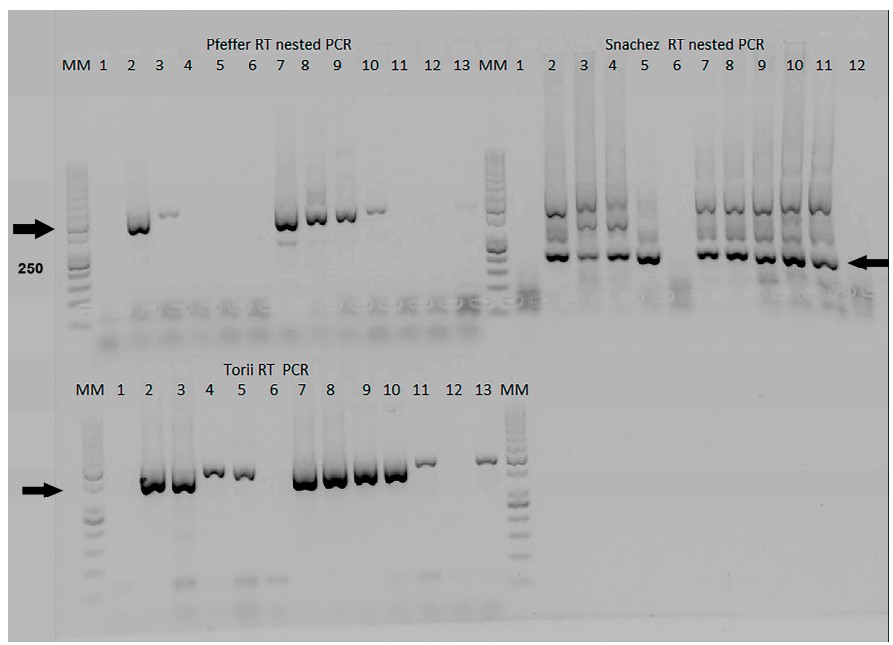

**Figure 3.** Agarose electrophoresis of RT-PCR products obtained using Pfeffer, Sanchez, and Torii methods. Lane 1 negative control (no virus), Lane 2—VEEV $10^7$, Lane 3—VEEV $10^6$, Lane 4—VEEV $10^5$, Lane 5—VEEV $10^4$, Lane 6—VEEV $10^3$, Lane 7—EEEV/Sinv $10^7$, Lane 8—EEEV/Sinv $10^6$, Lane 9—EEEV/Sinv $10^5$, Lane 10—EEEV/Sinv $10^4$, Lane 11—EEEV/Sinv $10^3$, Lane 12—VEEV $10^3$, Lane 13—EEEV/Sinv $10^3$.

The one-step Pfeffer RT-PCR reliably detected VEEV at concentrations of $10^7$ and $10^6$ PFU/mL, but not at lower concentrations, and also detected the EEEV/Sinv concentration from $10^7$ PFU/mL to EEEV/Sinv $10^3$ PFU/mL. The Sanchez-nested RT-PCR successfully detected all dilutions of both viruses, except for both replicates of VEEV$10^3$ PFU/mL. Sample 13 from the nested Sánchez RT PCR EEEV/Sinv$10^3$ PFU/mL was not dispensed in the gel. No statistical differences were observed in this case between these methods $p = 0.08$ by McNemar, and the Kappa agreement was considered a moderate level of 0.5.

The Torii RT-PCR detected the same dilutions as the Sanchez-nested RT-PCR. No statistical significance was observed when Sanchez and Torii tests were compared, and the Kappa agreement was perfect.

Considering the sensitivity and specificity of each Universal PCR method for detecting Alphavirus, as well as the time and resources required to complete each protocol, the Torii one-step RT-PCR [6] has been selected for the next phase.

### 3.3. Brain Tissue Samples

No differences were observed when testing 1 µL or 5 µL of extracted RNA from control dilutions which were kept frozen for more than 3 years. Figure S1A shows the amplification results when 1 µL of RNA was used in the Master mix, while Figure S1B shows the result of using 5 µL of RNA. The Molecular marker (MM) 50 bp GeneRuler™ ladder was used.

Histological analysis of several brain regions of sample LSE9010-15 revealed severe non-suppurative encephalitis, characterized by moderate to severe perivascular lymphoplasmacytic infiltrates and moderate gliosis in those locations, consistent with a viral infection. In Figure 4A, it was observed that DNA amplicons were not obtained in all of the analyzed brain locations. Instead, faint DNA bands were visible in different regions of the

brain of this sample, while the strongest band was observed in the hippocampus region, Figure 4A.

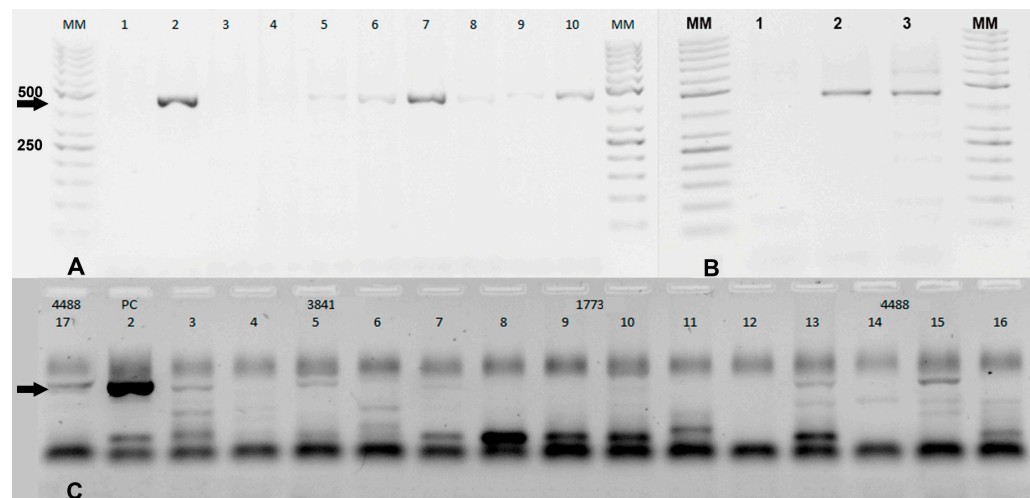

**Figure 4.** Evaluation of different brain locations in four equines exhibiting nervous signs. (**A**), Lane 1: Negative control (water- no virus), Lane 2—VEEV $7 \times 10^3$ PFU/mL, Lane 3 Medulla oblongata, Lane 4—Cerebellum, Lane 5—Basal nuclei, Lane 6—Thalamus, Lane 7—Hippocampus, Lane 8—Occipital cortex, Lane 9—Frontal colliculi, Lane 10—Spinal cord. (**B**), Lane 1 negative control (water-no virus), Lane 2—VEEV $7 \times 10^3$ PFU/mL, Lane 3—extraction pool of the hippocampus, spinal cord, and thalamus tissues from sample LSE9010-15. (**C**), Lane 2 VEEV $7 \times 10^3$ PFU/mL, Lanes 3 to 7 (sample LSE3841-2021), Lanes 8 to 12 LSE1773 (2021) and Lanes 13 to 17 LSE4488 (2020), each Lane corresponds to the following brain locations, basal nuclei, thalamus, hippocampus, frontal colliculi, and spinal cord.

In Figure 4B, a distinct band similar to the band produced by the positive control VEEV $7 \times 10^3$ PFU/mL can be observed in Lane 3 of the gel. Lane 3 represents a pool of frozen tissues extracted from the hippocampus, spinal cord, thalamus, and basal nuclei. It is important to note that this pooled sample corresponds to the same extraction samples used in Figure 4A, where DNA amplicons were observed in different brain regions. The presence of a clear band at the same position as the positive control indicates that the viral load in this pooled sample is comparable to the limit of detection (LOD) for VEEV using this specific method.

In Figure 4C, the amplification bands of the four positive equine samples detected in this study out of the 30 equine samples are shown. A molecular marker (MM) of 50 bp was used for size determination, and the expected amplicon product at 460 bp is indicated by arrows.

This figure presents the results of three individual brain samples; the LSE3841 sample (collected in 2021) displayed bands in Lane 3 (basal nuclei location) and Lane 5 (hippocampus). Sample LSE1773 brain (collected 2021) showed a faint band in Lane 10 (hippocampus). Finally, the brain sample LSE4488 (collected in 2020) exhibited bands in Lane 13 (basal nuclei), Lane 15 (hippocampus), and Lane 17 (spinal cord). These results indicate the variable presence of viral RNA in specific brain regions of the positive equine samples, demonstrating the viral infection in those regions.

Out of the four samples found positive for Alphavirus, only two samples were confirmed to be VEEV subtype IE by sequencing. These two samples were isolated in the northern region of Costa Rica.

One of the positive samples, LSE4488 (ON840547 GenBank ID), exhibited a 98.58% similarity to the sequence KC344441, which was isolated in Nicaragua in 1968. Additionally, it displayed a 98.10% of identity to the sequence of sample LSE9010-15 (MK796243) [15], also amplified in this study with Torii RT-PCR.

None of the brains collected from bovines exhibiting signs of encephalitis tested positive for Alphavirus when 5 μL of the extractions were used in the Torii RT-PCR method.

## 4. Discussion

Three of the RT-PCR nested and semi-nested assays were validated against several species of the Alphavirus genus [2–4], and the Torii RT-PCR was able to detect a new Alphavirus species in Zambia named Mwinilunga virus [5]. The Grywna et al. protocol was validated using 10 Alphaviruses and other RNA viruses. On the other hand, the Pfeffer et al. protocol was validated using 30 viruses, which included strains from all six VEEV subtypes. The Sanchez et al. protocol was validated using 12 viruses. Our intention was not to revalidate these methods but rather to verify and evaluate the performance of the assay under our specific laboratory conditions to ensure its accuracy and reliability. Factors such as the design of primers, the selection of target genes, and the conservation of the amplified regions can all impact the sensitivity and specificity of the assay. Additionally, variations in equipment, reaction volumes, and reagents used can also affect the assay's performance. This thorough evaluation helps to ensure that the assay is suitable for the intended purpose of detecting Alphavirus in the brain tissue of animals with encephalitis, providing accurate and reliable results.

It is interesting to note that three of the four Universal RT-PCR assays use the NSP4 gene as a target. In the case of the Grywna test, the limit of detection (LOD) corresponds to $10^4$ PFU/mL, ranging from 5 to 100 RNA copies per reaction across all Alphavirus species [9]. Meanwhile, Sanchez's LOD was $5 \times 10^6$ PFU/mL for EEEV in the first RT-PCR and $5 \times 10^2$ PFU/mL in the nested PCR, corresponding to 1 copy per reaction [7]. No LOD information is available in the Torii manuscript [6]. While the Pfeffer et al. method targeted the NSP1 gene for its RT-PCR, the LOD was $1.2 \times 10^3$ PFU/mL of VEE strain for the first RT-PCR and 1.2 PFU/mL for the semi-nested PCR [8]. The final reaction volume used in the comparison of the four Universal RT-PCR was 12.5 μL, while in the Pfeffer protocol, it was 100 μL, 50 μL in Grywna and Sanchez protocols, and 15 μL in the Torii assay. Of the RT-PCRs, only the Sanchez protocol shows several unspecific bands when positive control samples were used. In this nested RT-PCR, the band corresponding to the 195 bp band in the Chimera EEE/Sinv was confirmed as Sindbis virus by sequencing. Recombinant Sindbis virus (SINV)/VEEV and SINV/EEEV constructs have been developed to express the immunogenic structural proteins derived from VEEV or EEEV within the less virulent SINV backbone, which include the NSPs. We use this chimeric virus because this approach allowed us to prepare control stocks of the chimeric virus while adhering to the appropriate biosafety guidelines and minimizing potential risks associated with handling VEEV or EEEV during their inoculation in Vero cells.

We observed that VEEV did not amplify in all of the brain locations tested. In studies of Alphavirus replication in mouse brains, histopathological evidence of EEEV damage was detected in the cortex, hippocampus, and thalamus [16]. Conversely, VEEV viral replication in the brains of mice was multifocal and localized in association with brain capillaries [17]. In humans, EEEV produces lesions in the basal ganglia, thalamus, and cerebral cortex, visible by computed tomography (CT) or magnetic resonance imaging (MRI) [18]. In our study, we amplified VEEV RNA in the hippocampus, thalamus basal nuclei, and spinal cord of the equines. We were able to amplify pools of these locations from two positive equines, demonstrating the possibility of using these pools in VEEV diagnosis. Overall, these findings highlight the importance of selecting the correct sampling site in the brain for accurate diagnosis of viral infections. An interesting observation is the presence of nonspecific bands in the Torii RT-PCR when the brain tissue matrix and host DNA are included in the sample. This phenomenon can be attributed to the use of degenerate primers in the Torii et al. protocol. Degenerate primers contain nucleotide positions where multiple possibilities are allowed, increasing the chances of binding to nonspecific DNA sequences.

Despite the seropositive cases of VEEV in Costa Rica ranging from approximately 23% to 36% [10,11], only four out of the 30 equine brains evaluated in this study showed amplifi-

cation for Alphavirus. Furthermore, only two samples could be sequenced and confirmed as VEEV subtype IE due to the low viral load observed in the other two samples. The challenges of detecting RNA are due to its susceptibility to degradation by multiple freeze and thaw cycles or RNase enzymes present in the tissue. To address this, the volume of RNA extraction was increased to 5 μL in the RT-PCR. Alphaviruses can be diagnosed not only in tissues but also in blood, although viremia is generally detected in blood for only 3 to 6 days after infection [19]. We kept positive controls frozen for more than 3 years without significant loss of viral load. This is a valuable finding, indicating the stability of the viral material under the storage conditions used. However, it is important to note that despite this stability, the level of viral DNA amplification in some of the brain samples was very low. Several factors could potentially contribute to the poor amplification levels observed in certain brain samples. Despite stable viral load in positive controls, the brain samples might have experienced some degradation of viral genetic material during long-term storage. Factors like temperature fluctuations, repeated freeze-thaw cycles, and extended storage duration could contribute to DNA degradation. Brain tissues can contain substances that inhibit the PCR reaction, interfering with the amplification process and leading to poor results. A study used the detection of canine distemper virus (CDV) as an internal control to evaluate the extraction method, RNA degradation, and the presence of PCR inhibitors in bovine brain samples. CDV was detected in all the extracted samples when it was spiked directly into the extraction process but not in all the samples when CDV was spiked directly into the brain tissue sample before RNA extraction, possibly due to rapid CDV and RNA degradation in the brain tissue matrix [20]. The autophagy process, which is used by cells to maintain cellular health and homeostasis, may also contribute to the failure of viral RNA detection in tissue samples [21]. Another possible reason could be a low viral load, making it challenging to detect and amplify the viral DNA, especially if the PCR method used has a higher limit of detection. It is also possible that these samples were negative for Alphavirus, and another cause is responsible for producing the clinical signs.

We demonstrate that the Torii RT-PCR is well-suited for use in our laboratory conditions, with our reagents (master mix reaction volume, extraction methods), equipment (different thermocyclers, micropipettes), and workflow (one or two RT-PCR steps, different brain locations), it was also inexpensive, faster, sensitive, and specific in comparison with the nested RT-PCRs in the same conditions. It appears that the most suitable tissues for VEEV diagnosis in equines are the hippocampus, basal nuclei, and spinal cord. These specific tissue samples have shown positive amplification of VEEV RNA in our study, indicating their potential for accurate diagnosis of VEEV infection in equines. Finally, we were able to sequence two samples; according to the Basic Local Alignment Search Tool Blast [6] (National Institute of Health, Bethesda, MD, USA), both viruses belong to the VEEV subtype IE confirming the endemicity of this subtype at least in the northern provinces of Costa Rica. The presence of this virus highlights the importance of ongoing surveillance and understanding of these viruses to implement effective disease control measures under the one health protocol.

**Supplementary Materials:** The following supporting information can be downloaded at: https://www.mdpi.com/article/10.3390/zoonoticdis3030018/s1, Figure S1: Comparison between 1 μL and 5 μL of the extracted RNA and different extraction protocols.; Table S1: Retro transcription protocol; Table S2. First phase PCR two steps format of the Grywna, Pfeffer, and Sanchez protocols; Table S3. Phase II, RT-PCR one-step format of the Pfeffer, Sanchez, and Torii protocols.

**Author Contributions:** Conceptualization, B.L.; methodology, B.L. and J.E.-M.; formal analysis, B.L. and J.E.-M.; investigation, B.L.; writing—original draft preparation, B.L.; writing—review and editing, B.L. and J.E.-M.; supervision, C.J.; funding acquisition, C.J. All authors have read and agreed to the published version of the manuscript.

**Funding:** Consejo Nacional de Rectores (agreement-VI-177-2012, and agreement-VI-270-2017), University-government cooperation Ministerio de Planificación- Consejo Nacional de Rectores 2017, and PROMOTORA COSTARRICENSE DE INNOVACIÓN E INVESTIGACIÓN, Contrato de Incentivos N° FI-231B-17.

**Institutional Review Board Statement:** Not applicable.

**Informed Consent Statement:** Not applicable.

**Data Availability Statement:** The raw data from this study can be obtained by requesting it from the corresponding authors.

**Acknowledgments:** Sandra Sandra López Vergès, Dpt. of Research in Virology and Biotechnology Gorgas Memorial Research Institute for Health Studies, Panama, and Heidi Wood (Chief, Zoonotic Diseases and Special Pathogens Division), Kai Makowski, at National Microbiology Laboratory, Public Health Agency of Canada/Government of Canada, for supplying positive Alphavirus controls which allow this study, Gabriel Gonzalez and Arturo Molina for reviewing the manuscript, and especially, Tracy L Sturgill, for also reviewing the manuscript and for her comments.

**Conflicts of Interest:** The authors declare no conflict of interest.

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
