# Peer review of "Detection of Venezuelan Equine Encephalitis Virus from Brain Samples of Equines with Encephalitis"

_zoonoticdis, doi:10.3390/zoonoticdis3030018_

Round 1
Reviewer 1 Report
Comments and Suggestions for Authors
I followed the studies that were conducted but the Results require more information. The authors should introduce what they plan to do and then state what they did.
The figure legends should stand alone and include why there are arrows. Each figure should have a title and then explain what it is. For example. Figure 3. Agarose electrophoresis of RT-PCR products obtained using Pfeffer, Sanchez and Torii methods. Lanes for each method correspond to the same sample. Samples are as follows; Lane 1+negative control (no virus), Lane 2-VEEV 107PFU/ml, etc
Overall the revision requires careful editing and inclusion of information or this paper will not be useful to anyone.
Comments on the Quality of English Languageas above- extensive edits need to be made to English
Author Response
Reviewer 1
Comments and Suggestions for Authors
I followed the studies that were conducted but the Results require more information. The authors should introduce what they plan to do and then state what they did.
R/ Thank you for your comment, we add the following brief explanation to introduce in the result part, please see Lines 156-158:
Our primary aim was to enhance the diagnostic surveillance of animals exhibiting encephalitis signs. Throughout this investigation, we evaluated different protocol formats to determine the most effective method for our study.
The figure legends should stand alone and include why there are arrows. Each figure should have a title and then explain what it is. For example. Figure 3. Agarose electrophoresis of RT-PCR products obtained using Pfeffer, Sanchez and Torii methods. Lanes for each method correspond to the same sample. Samples are as follows; Lane 1+negative control (no virus), Lane 2-VEEV 107PFU/ml, etc
R/ Thank you very much for your recommendation, the Figure titles were modified, and the information about the arrows was included in the result section to reduce the text under the figure as was required by another reviewer, please see the following modifications in Figures 2-4 and S1.
Lines 246-247: Figure 2. Comparison of the three universal nested RT-PCR assays using the two-step format.
Lines 248-251:
Lane 1 negative control (no virus). Lanes 2 to 11 show VEEV positive controls, with varying concentrations (107, 105, 104, 104, 5x103, 5x103, 2.5x103, 2.5x103, 103, 103) expressed in PFU/mL. Lanes 12 to 21 depict the EEEV/Sinv Positive control, with the following concentrations (106, 105, 104, 104, 5x103, 5x103, 2.5x103, 5x103, 103, 103) expressed in PFU/mL.
Lines 256-257: Figure 3. Agarose electrophoresis of RT-PCR products obtained using Pfeffer, Sanchez and Torii methods.
Lines 258-261:
Lane 1 negative control (no virus), Lane 2- VEEV 107, Lane 3-VEEV 106, Lane 4-VEEV 105, Lane 5-VEEV 104, Lane 6-VEEV 103, Lane 7-EEEV/Sinv 107, Lane 8-EEEV/Sinv 106, Lane 9-EEEV/Sinv 105, Lane 10-EEEV/Sinv 104, Lane 11-EEEV/Sinv 103, Lane 12-VEEV 103, Lane 13-EEEV/Sinv 103.
Line 282: Figure 4. Evaluation of different brain locations in four equines exhibiting nervous signs.
Lines 283-292:
Figure 4A, Lane 1: Negative control (water- no virus), Lane 2-VEEV 7x103 PFU/mL, Lane 3 Medulla oblongata, Lane 4-Cerebellum, Lane 5- Basal nuclei, Lane 6- Thalamus, Lane 7- Hippocampus, Lane 8- Occipital cortex, Lane 9- Frontal colliculi, Lane 10-Spinal cord.
Figure 4B, Lane 1 negative control (water-no virus), Lane 2-VEEV 7x103 PFU/mL, Lane 3- extraction pool of the hippocampus, spinal cord, and thalamus tissues from sample LSE9010-15.
Figure 4C, Lane 2 VEEV 7x103 PFU/mL, Lanes 3 to 7 (sample LSE3841-2021), Lanes 8 to 12 LSE1773 (2021) and Lanes 13 to 17 LSE4488 (2020), each lane corresponds to the following brain locations, basal nuclei, thalamus, hippocampus, frontal colliculi, and spinal cord.
Lines 420-423: Lane 1 negative control (water-no virus), Lane 2-VEEV 7x105 PFU/mL, Lane 3-EEEV/Sinv 1.2x105 PFU/mL, Lane 4-VEEV 5x107 PFU/mL, Lane 5-VEEV 5x104 PFU/mL, Lane 6-VEEV 5x103 PFU/mL, Lane 7-EEEV/Sinv 4.5x105 PFU/mL, Lane 8-EEEV/Sinv 4.5x104 PFU/mL, Lane 9-EEEV/Sinv 4.5x103 PFU/mL, Lane 10-EEEV/Sinv 4.5x102 PFU/mL.
Overall the revision requires careful editing and inclusion of information or this paper will not be useful to anyone.
R/ Editing and text revision was done, please indicate which information should be added, thank you for your comment.
Reviewer 2 Report
Comments and Suggestions for Authors
The manuscript describes a study on Alphavirus presence in bovines and equines with encephalitis symptoms. Four Universal RT-PCR methods were compared, and the most suitable one was selected for the laboratory conditions. None of the bovine brains tested positive for Alphavirus, but among the 30 equine samples, only four were positive using the selected method. Sanger sequencing method confirmed that only two of these samples were of the Venezuelan equine encephalitis virus subtype IE. These samples were isolated in the northern region of Costa Rica. In my opinion the article is general well written, however the organization of the study makes it somewhat challenging to read. I have few very minor suggestions for the authors and one major which should be improved/ consider before publishing the paper:
1. Line 65: in my opinion it should be used“ positive control” instead of “control sample”
2. Line 210 Torii, it should be adding reference
3. The figure captions are too long, making it difficult to interpret the figures. Please shorten them and remove any comments that should be included in the Results section.
4. In my opinion, the article has a significant flaw: clinical samples should be tested using all methods to make a meaningful comparison of their effectiveness. In its current form, the manuscript is incomplete. Therefore, I propose conducting tests on samples using all methods and comparing them. Additionally, basic statistics on the results obtained from diluted samples would also be appropriate.
Comments on the Quality of English LanguageModerate editing of English language required.
Author Response
Reviewer 2
The manuscript describes a study on Alphavirus presence in bovines and equines with encephalitis symptoms. Four Universal RT-PCR methods were compared, and the most suitable one was selected for the laboratory conditions. None of the bovine brains tested positive for Alphavirus, but among the 30 equine samples, only four were positive using the selected method. Sanger sequencing method confirmed that only two of these samples were of the Venezuelan equine encephalitis virus subtype IE. These samples were isolated in the northern region of Costa Rica. In my opinion the article is general well written, however the organization of the study makes it somewhat challenging to read. I have few very minor suggestions for the authors and one major which should be improved/ consider before publishing the paper:
- Line 65: in my opinion it should be used“ positive control” instead of “control sample”
R/ the text was modified as follows, line 66 Positive Controls were provided by the Gorgas Memorial Institute of Health.
- Line 210 Torii, it should be adding reference
R/ The reference was added, lines 218: the Torii one-step RT-PCR [6], has been selected for the next phase.
- The figure captions are too long, making it difficult to interpret the figures. Please shorten them and remove any comments that should be included in the Results section.
R/ The figure captions were reduced as much as possible, please see the modifications in the following
Lines 246-247: Figure 2. Comparison of the three universal nested RT-PCR assays using the two-step format.
Lines 248-251:
Lane 1 negative control (no virus). Lanes 2 to 11 show VEEV positive controls, with varying concentrations (107, 105, 104, 104, 5x103, 5x103, 2.5x103, 2.5x103, 103, 103) expressed in PFU/mL. Lanes 12 to 21 depict the EEEV/Sinv Positive control, with the following concentrations (106, 105, 104, 104, 5x103, 5x103, 2.5x103, 5x103, 103, 103) expressed in PFU/mL.
Lines 256-257: Figure 3. Agarose electrophoresis of RT-PCR products obtained using Pfeffer, Sanchez and Torii methods.
Lines 258-261:
Lane 1 negative control (no virus), Lane 2- VEEV 107, Lane 3-VEEV 106, Lane 4-VEEV 105, Lane 5-VEEV 104, Lane 6-VEEV 103, Lane 7-EEEV/Sinv 107, Lane 8-EEEV/Sinv 106, Lane 9-EEEV/Sinv 105, Lane 10-EEEV/Sinv 104, Lane 11-EEEV/Sinv 103, Lane 12-VEEV 103, Lane 13-EEEV/Sinv 103.
Line 282: Figure 4. Evaluation of different brain locations in four equines exhibiting nervous signs.
Lines 283-292:
Figure 4A, Lane 1: Negative control (water- no virus), Lane 2-VEEV 7x103 PFU/mL, Lane 3 Medulla oblongata, Lane 4-Cerebellum, Lane 5- Basal nuclei, Lane 6- Thalamus, Lane 7- Hippocampus, Lane 8- Occipital cortex, Lane 9- Frontal colliculi, Lane 10-Spinal cord.
Figure 4B, Lane 1 negative control (water-no virus), Lane 2-VEEV 7x103 PFU/mL, Lane 3- extraction pool of the hippocampus, spinal cord, and thalamus tissues from sample LSE9010-15.
Figure 4C, Lane 2 VEEV 7x103 PFU/mL, Lanes 3 to 7 (sample LSE3841-2021), Lanes 8 to 12 LSE1773 (2021) and Lanes 13 to 17 LSE4488 (2020), each lane corresponds to the following brain locations, basal nuclei, thalamus, hippocampus, frontal colliculi, and spinal cord.
Lines 420-423: Lane 1 negative control (water-no virus), Lane 2-VEEV 7x105 PFU/mL, Lane 3-EEEV/Sinv 1.2x105 PFU/mL, Lane 4-VEEV 5x107 PFU/mL, Lane 5-VEEV 5x104 PFU/mL, Lane 6-VEEV 5x103 PFU/mL, Lane 7-EEEV/Sinv 4.5x105 PFU/mL, Lane 8-EEEV/Sinv 4.5x104 PFU/mL, Lane 9-EEEV/Sinv 4.5x103 PFU/mL, Lane 10-EEEV/Sinv 4.5x102 PFU/mL.
- In my opinion, the article has a significant flaw: clinical samples should be tested using all methods to make a meaningful comparison of their effectiveness. In its current form, the manuscript is incomplete. Therefore, I propose conducting tests on samples using all methods and comparing them. Additionally, basic statistics on the results obtained from diluted samples would also be appropriate.
R/ With all respect, our study aimed to determine the most sensitive, specific, and cost-effective method for detecting equine encephalitis alphavirus (EEV) in our laboratory conditions. This was crucial to avoid processing all samples with every RT-PCR test due to the significant associated costs. Our laboratory regularly receives brain samples from animals with nervous symptoms, which require accurate diagnoses of bovine spongiform encephalitis (BSE) in bovines and Rabies Virus.
Considering the importance of implementing differential methods for identifying new viral etiologies, we focused on evaluating various RT-PCR tests. The goal was to identify an efficient and low-cost test for detecting alphaviruses and determining the most suitable brain locations for conducting these tests, especially considering the constraints of our limited budget.
Our findings revealed that some of the methods were unable to detect VEEV in low concentrations, indicating that using these methods for processing all samples would be impractical, leading to a waste of time and resources. Consequently, we recommend employing specific methods based on the characteristics of the samples to optimize the diagnostic process for Equine Encephalitis Alphavirus detection and enhance the overall efficiency of our laboratory procedures.
The results of our study hold particular significance for developing countries like ours, providing valuable insights into efficient and cost-effective testing methods for alphaviruses. By sharing these findings through publication, we aim to contribute to the scientific community and support laboratories facing similar resource limitations in their efforts to combat viral diseases effectively.
On the other hand, we really appreciate your comment about basic statistics on the results obtained from diluted samples, we totally agree, please see the following test added to the manuscript in the lines indicate below.
Lines 113-114: The McNemar and Kappa tests were used to establish significant differences between the four methods evaluated during the first and second phases.
Lines 180-183: The Kappa agreement test returned a fair agreement level of 0.22 between the two methods. Furthermore, the differences observed between these two methods were found to be statistically significant according to the McNemar test (p= 0.0004).
Lines 189-192: In comparison to the Pfeffer test, the Sanchez-nested RT-PCR detected ten more positive controls. The Kappa agreement test also showed a fair agreement level of 0.22 between the two methods. Furthermore, the differences observed were found to be statistically significant according to the McNemar test (p= 0.0015).
Lines 208-209: No statistical differences were observed in this case between these methods p=0.08 by McNemar and the Kappa agreement was considered a moderate level of 0.5.
Lines 212-214: The Torii RT-PCR detected the same dilutions as the Sanchez-nested RT-PCR. No statistical significance was observed when Sanchez and Torii tests were compared and the Kappa agreement was perfect.
Comments on the Quality of English Language
Moderate editing of English language required.
Round 2
Reviewer 2 Report
Comments and Suggestions for Authors
Thank you for your answers. Especially the part concerning statistics greatly enriched the entire article.
In my opinion, the article in this form is ready to be published.